# Effect of Ipsilateral, Contralateral or Bilateral Repetitive Transcranial Magnetic Stimulation in Patients with Lateralized Tinnitus: A Placebo-Controlled Randomized Study

**DOI:** 10.3390/brainsci12060733

**Published:** 2022-06-02

**Authors:** Yi Li, Yong-Cong Shen, John J. Galvin, Ji-Sheng Liu, Duo-Duo Tao

**Affiliations:** 1Department of Ear, Nose, and Throat, Dushu Lake Hospital Affiliated of Soochow University, Suzhou 215000, China; 20195232187@stu.suda.edu.cn; 2Department of Ear, Nose, and Throat, The First Affiliated of Soochow University, Suzhou 215000, China; 20195232201@stu.suda.edu.cn; 3House Institute Foundation, Los Angeles, CA 90057, USA; jgalvin@hei.org; 4University Hospital Center of Tours, 37000 Tours, France; 5Department of Ear, Nose, and Throat, Shaanxi Provincial People’s Hospital, Xi’an 710068, China

**Keywords:** subjective tinnitus, lateralized tinnitus, repetitive transcranial magnetic stimulation, visual analogue scale

## Abstract

The relative benefit of ipsilateral, contralateral, and bilateral repetitive transcranial magnetic stimulation (rTMS) for tinnitus treatment remains unclear, especially for patients with lateralized tinnitus. In this study, we compared outcomes after 10 sessions of 1-Hz rTMS at 110% of resting motor threshold over a two-week period. In total, 104 right-handed patients with lateralized subjective tinnitus were randomly divided into four groups according to rTMS treatment: Left (*n* = 29), Right (*n* = 23), Bilateral (*n* = 30), and Sham stimulation (*n* = 22). Outcomes included estimates of tinnitus severity, psychological state, and psychoacoustic measures. Patients with left- or right-sided tinnitus were similarly distributed across treatment groups. There were no significant changes in outcome measures for the Right or Sham treatment groups. For the Left and Bilateral groups, tinnitus severity was significantly lower after treatment (*p* < 0.05). The reduction in tinnitus severity was largest for ipsilateral treatment in the Left group. The overall response rate was 56.1% for the Left group, 46.7% for the Bilateral group, 8.3% for the Right group, and 8.3% for the Sham group. For the Left and Bilateral groups, the response rate was larger for patients with left- than right-sided tinnitus. Changes in tinnitus severity were best predicted by changes in anxiety, depression, and the loudness of the tinnitus. The results suggests that rTMS on the left temporoparietal cortex is more effective for patients with left-sided than with right-sided tinnitus.

## 1. Introduction

Subjective tinnitus is characterized by the perception of sound in the absence of a corresponding acoustic stimulus [1]. The pathophysiologic mechanisms of tinnitus are not fully understood [2]. Current available management strategies for tinnitus are diverse, and include pharmacotherapy [3], cognitive-behavioral therapies [4] and sound therapies [5]. However, these approaches have shown mixed results in reducing tinnitus severity [6].

Repetitive transcranial magnetic stimulation (rTMS) is a non-invasive tool that is used to deliver electromagnetic pulses to the brain across the scalp via a coil [7]. Low-frequency rTMS applied to the left temporoparietal cortex has been proposed as a novel and effective treatment for chronic tinnitus [8,9,10,11]. For patients with lateralized tinnitus, there is some uncertainty regarding the location of rTMS, relative to the tinnitus side. Marcondes et al. [12] treated 10 tinnitus patients with 1-Hz rTMS on the left temporoparietal cortex and 10 patients with sham stimulation; a significant reduction in visual analog scale (VAS) scores of tinnitus severity was observed immediately after treatment only for the active rTMS group. Frank et al. [13] reported that rTMS significantly reduced tinnitus severity in patients with left-sided tinnitus, but not with right-sided tinnitus. Khedr et al. [14] found that contralateral rTMS stimulation had a greater effect than ipsilateral stimulation, relative to the side of unilateral tinnitus. Kim et al. [15] found no significant difference between contralateral and ipsilateral treatment after 1 month of rTMS, relative to the side of unilateral tinnitus. These previous studies show mixed results regarding the optimal treatment side relative to the side of unilateral tinnitus. The efficacy of bilateral rTMS for patients with lateralized tinnitus also remains unclear, as well as the relative benefits for ipsilateral, contralateral, and bilateral stimulation.

It is difficult to determine the benefit of rTMS beyond the placebo effect. Some studies have shown a significant benefit for real rTMS compared to sham stimulation [10,16,17], while others have not [18,19,20]. In a randomized controlled trial, Folmer et al. [10] observed a statistically higher response rate of tinnitus after rTMS treatment. Rossi et al. [16] compared active and sham rTMS in 14 patients with chronic tinnitus. After two weeks of treatment, the response rate in the active stimulation group was significantly higher than that in the sham stimulation group. In a meta review of randomized clinical trials, Soleimani et al. [17] reported a significant benefit of rTMS in terms of improved Tinnitus Handicap Inventory (THI) scores. However, other researchers have not shown a significant advantage of rTMS over sham treatment. Sahlsten et al. [20] found no significant difference in the reduction of tinnitus intensity, annoyance, and stress, or in THI scores between active and sham stimulation, although the percent of responders was higher for the active group than for the sham group. Hoekstra et al. [18] found no significant advantage for active rTMS over sham stimulation at any time point after treatment. Collectively, these studies indicate that the benefit of rTMS for tinnitus treatment remains uncertain.

The aims of the present study were: (1) to explore the most effective strategy for rTMS treatment in terms of treatment side, (2) to explore interactions between treatment side and laterality of tinnitus, and (3) to determine the benefit of rTMS compared to the placebo effect. The primary outcome measures were tinnitus severity in terms of VAS and THI scores. Secondary outcome measures included anxiety and depression scales, pitch matches to tinnitus, and loudness matches to tinnitus.

## 2. Materials and Methods

### 2.1. Participants

This study was conducted between July 2019 and August 2020. All procedures for recruitment, informed consent, and the conduct of the study adhered to the requirements of the Institutional Review Board at the First Affiliated Hospital of Soochow University (#2019097). The selection criteria were as follows: (1) adult patients (≥18 <70 years old), (2) subjective tinnitus as the first clinical complaint, (3) no effect of previous routine therapies (e.g., glucocorticoids), and (4) the ability to independently complete all of the assigned questionnaires. The exclusion criteria were as follows: (1) family history of epilepsy, metal implants in the head or body, pacemakers, acute brain trauma, cerebral hemorrhage, intracranial infection or increased intracranial pressure or pregnancy, (2) objective tinnitus, (3) Meniere’s disease or sudden deafness, (4) patients with communication disorders or the inability to communicate or complete the questionnaires. All participants had lateralized tonal tinnitus. Hearing status, in terms of pure-tone average (PTA) thresholds across 0.5, 1, 2, and 4 kHz, was not part of the inclusion or exclusion criteria. A total of 104 patients were recruited for the study and were randomly assigned to the Left (*n* = 29), Right (*n* = 23), Bilateral (*n* = 30), and Sham (*n* = 22), treatment groups (see Table 1 for demographic information). All patients were right-handed. Note that in China, the prevalence of left-handedness is less than 1% [21,22].

While there was no significant difference in age across the groups (*p* = 0.151), age at testing was lower for the Bilateral (mean = 36.4 yrs; range = 18–61) than for the Left (mean = 44.2 yrs; range = 25–66), Right (mean = 40.8 yrs; range = 20–67), or Sham groups (mean = 39.9 yrs; range = 30–58). PTA thresholds were significantly lower for the Bilateral (mean = 22.2 dB HL; range = 5–50) than for the Right (mean = 38.9 dB HL; range = 10–75) or Sham groups (mean = 34.3 dB HL; range = 20–95). The duration of tinnitus was significantly longer for the Left (mean = 3.2 yrs; range = 0.08–10) than for the Bilateral group (mean = 1.6 yrs; range = 0.17–7).

### 2.2. Outcome Measures

Outcomes were measured before and after 10 sessions of 1-Hz, rTMS at 110% of resting motor threshold (RMT) over a two-week period. Outcome measures before and after treatment included:VAS score of tinnitus severity [23]. Patients were asked to mark their tinnitus severity on a 10-cm line, where 0 = no tinnitus and 10 = worst tinnitus imaginable. Cartoon expressions (e.g., smile, neutral, pain, extreme pain, etc.) were distributed above the line to illustrate the range of tinnitus severity. VAS scores were compared pre- and post-treatment. A response to treatment was considered when VAS scores were reduced by 10% or more [24].THI score [25,26]. The THI contains 25 questions with 3 response choices (Yes—4 points; Sometimes—2 points; No—0 points), with a score of 100 indicating maximum tinnitus severity. A response to treatment was considered a reduction of 6 points or more in THI scores [27,28].Hospital Anxiety and Depression Scale (HADS) [29]. Patient mood was characterized using the HADS questionnaire, which includes two subscales consisting of 7 questions each to assess anxiety (HADS-A) and depression (HADS-D). Eight or more points was considered to be clinically significant anxiety or depression.Pitch matching was performed using pure-tone stimuli delivered over headphones; stimuli were presented at 10 dB sensation level (SL) relative to hearing threshold. During testing, two sounds with different frequencies (e.g., 1 kHz and 8 kHz) were presented to the ear contralateral to the tinnitus, and the patient indicated which sound was closer to the pitch of the tinnitus. The frequencies were adjusted according to patient response, continuously narrowing the range until the frequency that best matched the pitch of the tinnitus was found.After identifying the frequency that best matched the pitch of the tinnitus, the loudness of the tinnitus was estimated. The stimulus level in the contralateral ear was adjusted to match the level of the tinnitus (5 dB initial step size, 1 dB final step size). This measurement of tinnitus was performed twice, with a short break between test runs. The final adjusted level was averaged across both runs and was expressed in terms of dB sensation level (SL), relative to the hearing threshold of the pitch-matched frequency.

### 2.3. rTMS

The RMT was first measured using a CCY-I transcranial magnetic stimulator (YRD CCY-I, Yiruide Company, Wuhan, China), which had a figure-of-8-shaped coil with a maximum external diameter of 92 mm and a magnetic field strength of 1.5 T. The left motor cortex M1 was stimulated using a single pulse. The activity of the abductor pollicis longus was recorded by electromyography (EMG). Patients were seated in a comfortable treatment chair. The RMT was defined as the lowest stimulator output intensity capable of inducing motor evoked potentials (MEPs) of at least 50 µV (peak-to-peak amplitude) in the relaxed state for at least 5 of 10 consecutive trials [30].

For the Left, Right, and Bilateral treatment groups, rTMS was performed using the same CYY-I transcranial magnetic stimulator; the stimulator is approved for tinnitus treatment in China. Using the International EEG System as an anatomical reference for rTMS, the location of the stimulation point was marked with a pen on the surface of the participant. The physiological condition and vital signs of the patients were closely observed during treatment. For the Left group, temporo-parietal stimulation was delivered from a single coil which was positioned on the midline between T3 (left temporal midpoint) and P3 (left parietal midpoint) regions. Stimulation consisted of a train of 10 biphasic pulses (one every second), followed by two skipped pulses and then 10 more pulses, two skipped pulses, etc. The 2-s rest was implemented to reduce the possibility of epilepsy. In all, there were 100 repeating sequences totaling 1000 pulses. For the Right group, the stimulation paradigm was the same, except that the coil was positioned on the midline between T4 (right temporal midpoint) and P4 (right parietal midpoint) regions. For the Bilateral group, temporo-parietal stimulation was simultaneously delivered from two coils that were positioned on the midline between the T3 (left temporal midpoint) and P3 (left parietal midpoint) regions and on the midline between the T4 (right temporal midpoint) and P4 (right parietal midpoint); as such, there were 200 repeating sequences totaling 2000 pulses. For the Sham group, participants received the same treatment as the Left group, but the stimulation coil was tilted away from the skull by 45° with one wing touching the skull to induce skin sensations without inducing magnetic stimulation, as in Landgrebe et al. [19]. For all groups, treatment was performed over 10 subsequent working days, 5 consecutive days over a two-week period).

### 2.4. Statical Analyses

Data analyses were performed using IBM SPSS (version 22.0; IBM, Armonk, NY, USA). For all analyses, significance was *p* < 0.05.

Demographic data were analyzed using chi-square or one-way analysis of variance (ANOVA), as appropriate linear mixed model (LMM) analyses were performed to compare pre- and post-treatment outcomes across groups and across patients with left- and right-sided tinnitus. Categorical fixed effects included group (Left, Right, Bilateral, Sham), tinnitus side (left, right), and treatment (pre, post); all factors of interest were included in the analysis. Participant was a random effect (intercept) for all LMMs. A maximum likelihood model was used for the LMMs. Pairwise comparisons were performed with Bonferroni correction.

Pearson correlation analyses were used to compare pre-treatment outcome measures to one another (to observe co-linearity) and to demographic variables; Bonferroni correction was applied for multiple comparisons.

Forward stepwise regression was used to identify predictors of tinnitus severity, as well as post-treatment changes in tinnitus severity. Response rates were analyzed using Mann–Whitney analyses.

## 3. Results

The complete dataset and patient demographic information can be found in Appendix A. Table 2 shows mean values of all outcome measures for each treatment group with left- or right-sided tinnitus.

### 3.1. Tinnitus Severity

Figure 1 shows post-treatment VAS scores as a function of pre-treatment scores. LMM analysis was performed on the VAS score data, with treatment group (Left, Right, Bilateral, Sham), tinnitus side (left, right), and test (pre, post) as fixed effects, and patient as a random effect. A significant effect was observed for only for test [F (1, 104) = 34.0, *p* < 0.001], and there was a significant interaction among treatment group, tinnitus side, and test [F (3, 104) = 2.8, *p* < 0.045]. Post-hoc Bonferroni pairwise comparisons showed that rTMS significantly reduced VAS scores for the Left group with left- or right-sided tinnitus (*p* < 0.05 for both comparisons), and for the Bilateral group with left-sided tinnitus (*p* < 0.05). There was no significant effect of rTMS for the Right or Sham groups, and no significant effect for the Bilateral group with right-sided tinnitus.

Figure 2 shows post-treatment THI scores as a function of pre-treatment scores. LMM analysis showed no significant effects for treatment group, tinnitus side, or test (*p* > 0.05 for all comparisons).

### 3.2. HADS Scores

Figure 3 shows post-treatment HADS-A scores as a function of pre-treatment scores. LMM analysis was performed on the HADS-A score data, with treatment group, tinnitus side, and test as fixed effects, and patient as a random effect. A significant effect was observed for test [F (1, 104) = 18.9, *p* < 0.001]. Post-hoc Bonferroni pairwise comparisons showed that rTMS significantly reduced HADS-A scores for the Left treatment group with left- (*p* = 0.014) and right-sided tinnitus (*p* = 0.002), and for the Bilateral group with right-sided tinnitus (*p* = 0.012). There were no significant effects for the Right or Sham groups.

Figure 4 shows post-treatment HADS-D scores as a function of pre-treatment scores. LMM analysis showed no significant effects for treatment, tinnitus side, or test (*p* > 0.05 for all comparisons).

### 3.3. Psychoacoustic Measures

Figure 5 shows post-treatment tinnitus pitch-matched frequencies as a function of pre-treatment measures. LMM analysis was performed on the pitch matching data, with treatment group, tinnitus side, and test as fixed effects and patient as a random effect. While there were no significant main effects, there was a significant interaction between treatment group and test [F (3, 104) = 2.7, *p* = 0.049]. Post-hoc Bonferroni pairwise comparisons showed that pitch-matched frequencies were significantly lower for the Left group with left-sided tinnitus after rTMS treatment (*p* = 0.001).

Figure 6 shows post-treatment tinnitus loudness matches at the pitch-matched frequency as a function of pre-treatment measures. LMM analysis was performed on the loudness matching data, with treatment group, tinnitus side, and test as fixed effects and patient as a random effect. A significant effect was observed for treatment group [F (3, 104) = 4.6, *p* = 0.004]. Post-hoc Bonferroni pairwise comparisons showed that loudness was significantly lower for the Sham group than for the Right (*p* = 0.042) and Bilateral groups (*p* = 0.003).

### 3.4. Pre- versus Post-Treatment

Table 3 shows the difference between pre- and post-training measures. LMM analysis was performed on the difference between pre- and post-treatment VAS scores, with treatment group and tinnitus side as fixed effects and participant as a random effect. Results showed a significant effect treatment group [F (3, 104) = 11.7, *p* < 0.001], but not for tinnitus side; there was a significant interaction [F (3, 104) = 2.8, *p* < 0.045]. Post-hoc Bonferroni pairwise comparisons showed that the reduction in VAS scores was significantly larger for the Left than for the Bilateral (*p* = 0.013), Right (*p* < 0.001), or Sham groups (*p* < 0.001), with no significant difference among the Bilateral, Right, and Sham groups. For the Bilateral group, the reduction in VAS scores was significantly larger for patients with left- than right-sided tinnitus (*p* = 0.007); there was no significant effect of tinnitus side for the Left, Right, or Sham groups. For patients with left-sided tinnitus, the reduction in VAS scores was significantly larger for the Left and Bilateral groups than for the Right and Sham groups (*p* < 0.05 for all comparisons), with no significant difference between the Left and Bilateral groups or between the Right and Sham groups. For patients with right-sided tinnitus, there was no significant difference in reduction of VAS scores across groups.

LMM analysis of changes in pitch-matching frequency showed a significant effect for treatment group [F (3, 104) = 2.7, *p* < 0.001], but not for tinnitus side. Post-hoc Bonferroni pairwise comparisons showed that the reduction in pitch-matched frequency was significantly larger for the Left than for the Right group (*p* = 0.045), with no significant differences among the remaining groups.

LMM analysis showed no significant effects of treatment group or tinnitus side on changes in THI scores, HADS-A scores, HADS-D scores, or tinnitus-matched loudness (*p* > 0.05 for all analyses.

Pearson correlations were used to observe significant associations among demographic data and pre-treatment outcome measures (Table 4); all patients were used for the analysis (*n* = 104). After Bonferroni correction for multiple comparisons (adjusted *p* = 0.00556), significant associations were observed between age at testing and duration of tinnitus, between VAS scores and THI, HADS-A, and HADS-D scores, between THI scores and HADS-A and HADS-D scores, and between HADS-A and HADS-D scores (*p* < 0.00556 in all cases).

Forward stepwise correlation analysis was performed on the pre-treatment data to identify variables that might predict tinnitus severity (Table 5). For VAS scores, age at testing, duration of tinnitus, PTA thresholds, THI scores, HADS-A scores, HADS-D scores, pitch-matched frequency, and loudness match were included in the model. Results showed that pre-treatment VAS scores were best predicted by THI scores, with the remaining variables not significantly contributing to the model. For THI scores, age at testing, duration of tinnitus, PTA thresholds, VAS scores HADS-A scores, HADS-D scores, pitch-matched frequency, and loudness match were included in the model. Results showed that pre-treatment THI scores were best predicted by a combination of VAS and HADS-A scores, with the remaining variables not significantly contributing to the model.

Forward stepwise correlation analysis was also performed on the change in post-training outcome measures (Table 3) to identify variables that might predict changes in post-treatment tinnitus severity; results are shown Table 6. For changes in VAS scores, age at testing, duration of tinnitus, PTA thresholds, THI scores, HADS-A scores, HADS-D scores, pitch-matched frequency, and loudness match were included in the model. Results showed that post-treatment changes in VAS scores were best predicted by a combination of changes in THI scores, HADS-A scores, and pitch-matched frequency to tinnitus, with the remaining variables not significantly contributing to the model. For THI scores, age at testing, duration of tinnitus, PTA thresholds, VAS scores HADS-A scores, HADS-D scores, pitch-matched frequency, and loudness match were included in the model. Results showed that post-treatment changes in THI scores were best predicted by a combination of HADS-A and HADS-D scores, with the remaining variables not significantly contributing to the model.

### 3.5. Treatment Response

Responders to rTMS were identified as having a reduction of 10% or more in VAS scores [24] or a reduction of 6 points or more in THI scores [27,28]. Figure 7 shows the percent of responders (gray) and non-responders (white) in terms of changes in VAS scores. For the Left group, the overall response rate was 56.1% (70.6 and 41.7% for left- and right-sided tinnitus, respectively). A Mann–Whitney test showed no significant difference in response rate between patients with left- or right-sided tinnitus (U = 82.0, *p* = 0.370). For the Bilateral group, the overall response rate was 46.7% (66.7 and 26.7% for left- and right-sided tinnitus, respectively). A Mann–Whitney test showed that the response rate was significantly higher for patients with left-sided than right-sided tinnitus (U = 54.5, *p* = 0.011). For the Right group, the overall response rate was 8.3% (0 and 16.7% for left- and right-sided tinnitus, respectively). A Mann–Whitney test showed no significant difference in response rate between patients with left- or right-sided tinnitus (U = 50.0, *p* = 0.103). For the Sham group, the overall response rate was 8.3% (0 and 16.7% for left- and right-sided tinnitus, respectively). A Mann–Whitney test showed no significant difference in response rate between patients with left- or right-sided tinnitus (U = 50.0, *p* = 0.209).

Figure 8 shows the percent of responders (gray) and non-responders (white) in terms of changes in THI scores. For the Left group, the overall response rate was 48.5% (47.1% and 50% for left- and right-sided tinnitus, respectively). A Mann–Whitney test showed no significant difference in response rate between patients with left- or right-sided tinnitus (U = 84.5, *p* = 0.450). For the Bilateral group, the overall response rate was 43.3% (46.7% and 40.0% for left- and right-sided tinnitus, respectively). A Mann–Whitney test showed no significant difference in response rate between patients with left- or right-sided tinnitus (U = 111.0, *p* = 0.967). For the Right group, the overall response rate was 21.5% (18.2% and 25.0% for left- and right-sided tinnitus, respectively). A Mann–Whitney test showed no significant difference in response rate between patients with left- or right-sided tinnitus (U = 63.3, *p* = 0.877). For the Sham group, the overall response rate was 8.3% (0% and 16.7% for left- and right-sided tinnitus, respectively). A Mann–Whitney test showed no significant difference in response rate between patients with left- or right-sided tinnitus (U = 52.0, *p* = 0.588).

## 4. Discussion

The was a significant benefit of rTMS in terms of reduction in VAS scores for the Left and Bilateral treatment groups, but not for the Right treatment group. Relatively few studies have applied rTMS to the right temporoparietal cortex. Previous studies have reported greater metabolic activity in the left than in the right primary auditory cortex in tinnitus patients [31,32]. Using PET imaging, Plewnia et al. [33] found greater activity in the left auditory cortex of chronic tinnitus patients, regardless of the side of symptoms. Smits et al. [34] performed functional magnetic resonance imaging (fMRI) of the brain pathway in 42 patients with tinnitus, including the inferior colliculus (IC), the medial geniculate body (MGB), the primary auditory cortex (A1, Heschl’s gyrus) and the secondary auditory cortex (A2, planum polare and planum temporale). Results showed a higher activation ratio in the left A2 and a lower activation ratio in the left IC. Sahlsten et al. [27] treated tinnitus patients using structural MRI-based navigated rTMS or electroencephalography (EEG)-based targeting of rTMS of the left temporoparietal cortex. The response rate did not significantly differ between the two treatment approaches, suggesting that the A1 may not be a critical stimulus site for rTMS treatment. The authors also found that rTMS treatment was more beneficial for patients with left-sided than with right-sided tinnitus, consistent with the present results. This further suggests that the target of stimulation may be in the secondary auditory cortex A2 rather than in the primary auditory cortex A1.

For the Left and Bilateral groups, the rTMS benefit was significantly larger in patients with left-sided than with right-sided tinnitus. Frank et al. [13] treated 194 tinnitus patients with rTMS on the left temporoparietal cortex and found that left stimulation had no significant effect on right-sided tinnitus. While the present response rate was larger for patients with left-sided tinnitus, the response rate for patients with right-sided tinnitus was substantial with Left (41.7%) or Bilateral treatment (26.7%). Note that in the present study, the pre-treatment VAS scores were significantly larger for the Left than for the Bilateral group (mean difference = 1.3 points), which may have affected treatment outcomes. In addition, the mean age at testing was lower for the Bilateral (36.4 yrs) than for the Left group (44.4 yrs), and PTA thresholds were lower for the Bilateral (22.2 dB HL) than for the Left group (34.3 dB HL), which may have contributed to the pattern of results. Still, the advantage observed for the Left treatment group suggests that the additional stimulation on the right side for the Bilateral treatment group provided little benefit.

No benefit was observed for the Sham treatment group. This finding was consistent with some previous studies that found no placebo effect [10,35,36,37], but not consistent with other studies that showed no significant difference between real and sham stimulation [18,19,20,38,39,40]. Discrepancies among studies regarding the placebo effect may be related to differences in patient characteristics (e.g., tinnitus duration, hearing loss level). Wang et al. [36] reported a large open-label study with 289 participants aimed at identifying the clinical predictors of tinnitus treatment efficacy. They found that significant suppression of tinnitus loudness (measured using a VAS) was correlated with tinnitus duration. De Ridder et al. [41] found that the maximal amount of suppression and best stimulation frequency depended on the tinnitus duration. They reported that the response to rTMS deteriorated with the progression of tinnitus, with greater efficacy during the first 1–3 years of tinnitus. In the present study, the mean duration of tinnitus was 3.2 years in the Left treatment group, much shorter than reported by Landgrebe et al. [19]. Moreover, the present change in VAS scores for tinnitus severity was not significantly related to duration of tinnitus (r = 0.11, *p* = 0.321). As such, the somewhat short duration of tinnitus in the present study may have increased response to rTMS.

In the present study, left temporoparietal stimulation also significantly reduced anxiety (HADS-A score) in tinnitus patients. Tinnitus was accompanied by anxiety in 65.5% of patients in the Left treatment group. Tinnitus is often accompanied by anxiety, depression and other adverse emotions [42,43]. In a randomized control study of tinnitus patients and healthy people, Mühlau et al. [44] found that the volume of gray matter in tinnitus patients was significantly reduced in the subcorpus callosum outside the auditory pathway, known to be involved with processing of auditory-induced unpleasant emotions. They suggested that the participation of the emotional region contributes to the perception of tinnitus. These observations have been confirmed by imaging studies [45,46] that suggest tinnitus may be a network system problem that involves higher-order cognitive cortical and limbic systems. It is possible that the therapeutic effect of rTMS observed in the present study may be related to improvement in mood (e.g., reduced HADS-A and HADS-D scores), rather than a specific rTMS effect. This is somewhat reflected by the forward linear regression analysis showing that reductions in THI scores were significantly predicted by a combination of the reduction in HADS-A and HADS-D scores. Note that stress was not tested in this study. Future studies should include instruments to measure stress in tinnitus patients, such as the Depression Anxiety and Stress Scale 21 (DASS-21) [47,48].

The advantage with non-navigated rTMS was not due to stimulation intensity, as there was no significant difference in RMT between the navigated and non-navigated group. The authors suggested that the stimulation site according to the 10–20 EEG system may have been more optimal, stimulating a wider brain area than with the more targeted navigated rTMS.

Significant associations were observed among pre-treatment measures of tinnitus severity and psychological state (Table 4). While there was a significant change in relative pitch match for the Left treatment group, there was no significant post-treatment change in loudness-matching for any of the treatment groups. This finding is not consistent with Sahlsten et al. [49], who treated 13 tinnitus patients and found that the loudness of tinnitus decreased significantly and the pitch of tinnitus changed in the majority of patients after rTMS treatment. Note that Sahlsten et al. [49] used navigated rTMS, while the present study did not. Navigated rTMS may have allowed for better localization of the target stimulation area. However, in a more recent study also using navigated rTMS, Sahlsten et al. [27] found that the loudness of tinnitus was reduced in both the rTMS and sham treatment groups. The difference in treatment outcomes across groups was not significant, suggesting that a placebo effect may have occurred. Few studies have found a significant relationship between tinnitus severity and the loudness or pitch of tinnitus [50,51,52,53]. These studies suggest that loudness is not a significant contributor to the perceived distress caused by tinnitus [54,55]. Factors other than loudness include duration of tinnitus and the psychological state of the patient [56,57]. Although tinnitus perceived at a greater loudness may be more annoying, it does not follow that softer tinnitus is any less of an issue for some patients. It is often the case that the perceived intensity of tinnitus does not determine patient response and distress [58].

There were several limitations to the present study. First, there was no follow-up after the 10 sessions of rTMS. Mennemeier et al. [59] suggested that patients who respond to initial rTMS treatment may need maintenance therapy every 3–6 weeks to gain long-term benefits. Longitudinal studies are needed determine the best protocol for rTMS stimulation over the long term, with and without maintenance therapy. Another weakness is that neither navigation nor imaging were used in this study, both of which can improve the efficacy of rTMS [60,61]. Noh et al. [62] found that tinnitus was similarly improved by 1 Hz-rTMS delivered over the left auditory cortex when an image-guided navigation system was used, or when defined as posterior to the T3–C3 line based on the 10–20 EEG System, as in Langguth et al. [30]. Sahlsten et al. [27] found no significant difference between navigated and non-navigated rTMS. While chronic tinnitus was significantly reduced in both groups, the treatment response was better in the non-navigated group in terms of reduced tinnitus intensity. The advantage with non-navigated rTMS was not due to stimulation intensity, as there was no significant difference in RMT between the navigated and non-navigated group. The authors suggested that the stimulation site according to the 10–20 EEG system may have been more optimal, stimulating a wider brain area than with the more targeted navigated rTMS.

## 5. Conclusions

The primary findings of the study were: (1) The reduction in tinnitus severity was greater with ipsilateral stimulation of the left temporoparietal cortex than with bilateral stimulation; (2) Treatment benefits were greater for patients with left-sided than with right-sided tinnitus; (3) No significant benefit was observed for stimulation of the right temporoparietal cortex; (4) rTMS was effective in reducing tinnitus severity beyond the placebo effect. The present results suggest that 1-Hz rTMS applied to the left temporoparietal cortex may reduce tinnitus severity, especially for patients with left-sided tinnitus.

## Figures and Tables

**Figure 1 brainsci-12-00733-f001:**
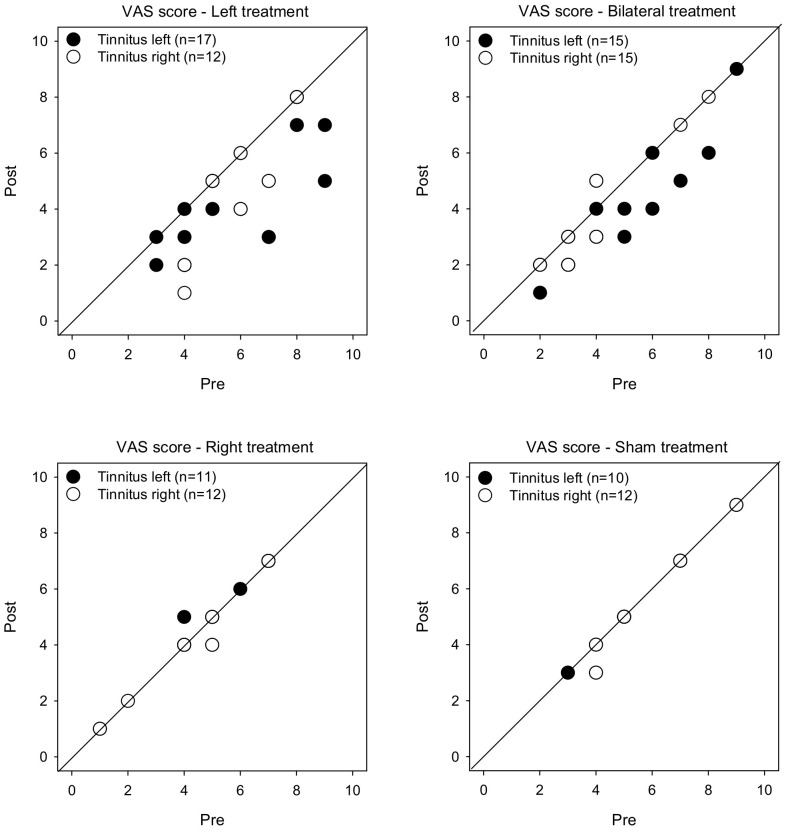
Post-treatment VAS scores as a function of pre-treatment scores for the Left, Right, Bilateral and Sham treatment groups. The filled and open symbols show data for patients with left- or right-sided tinnitus, respectively. Values below the diagonal line indicate reduced VAS scores after treatment.

**Figure 2 brainsci-12-00733-f002:**
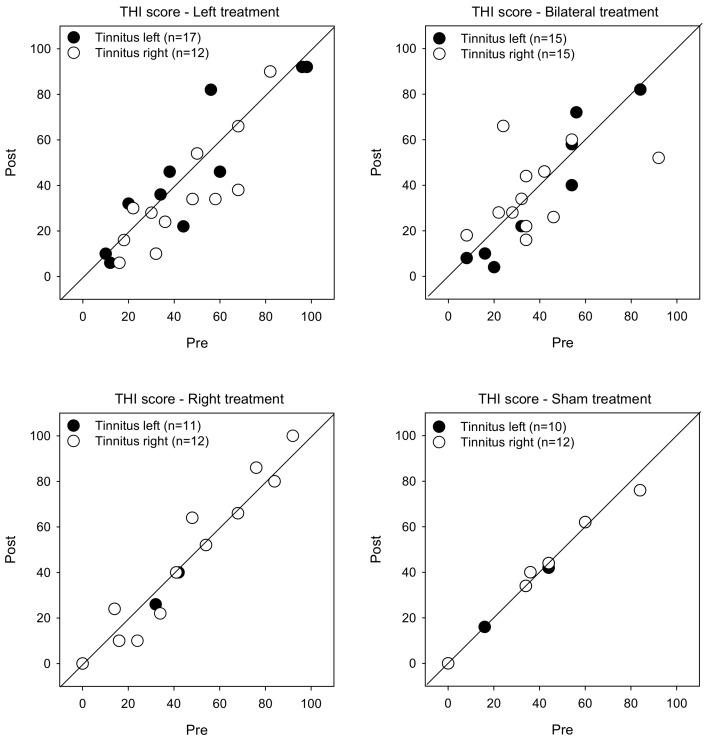
Post-treatment THI scores as a function of pre-treatment scores for the Left, Right, Bilateral and Sham treatment groups. The filled and open symbols show data for patients with left- or right-sided tinnitus, respectively. Values below the diagonal line indicate reduced THI scores after treatment.

**Figure 3 brainsci-12-00733-f003:**
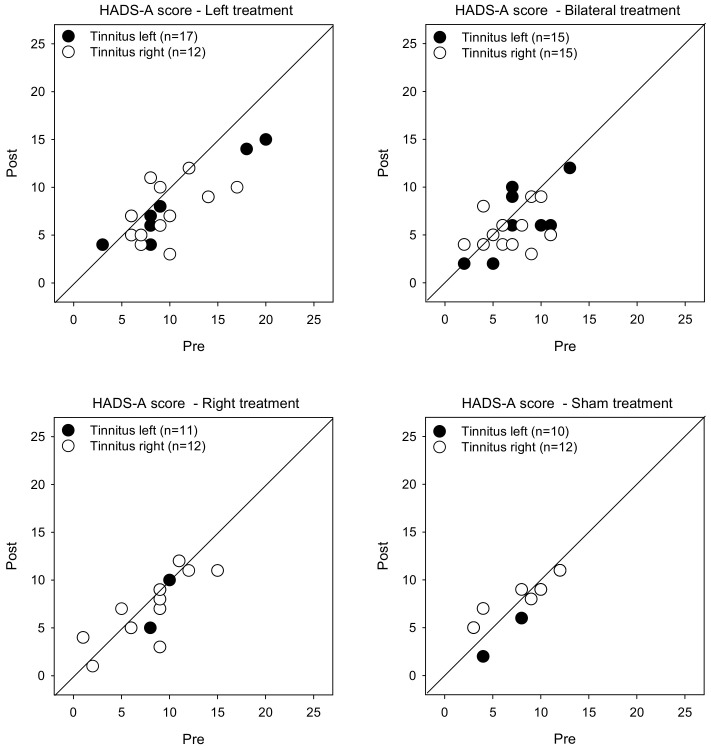
Post-treatment HADS-A scores as a function of pre-treatment scores for the Left, Right, Bilateral, and Sham treatment groups. The filled and open symbols show data for patients with left- or right-sided tinnitus, respectively. Values below the diagonal line indicate reduced anxiety after treatment.

**Figure 4 brainsci-12-00733-f004:**
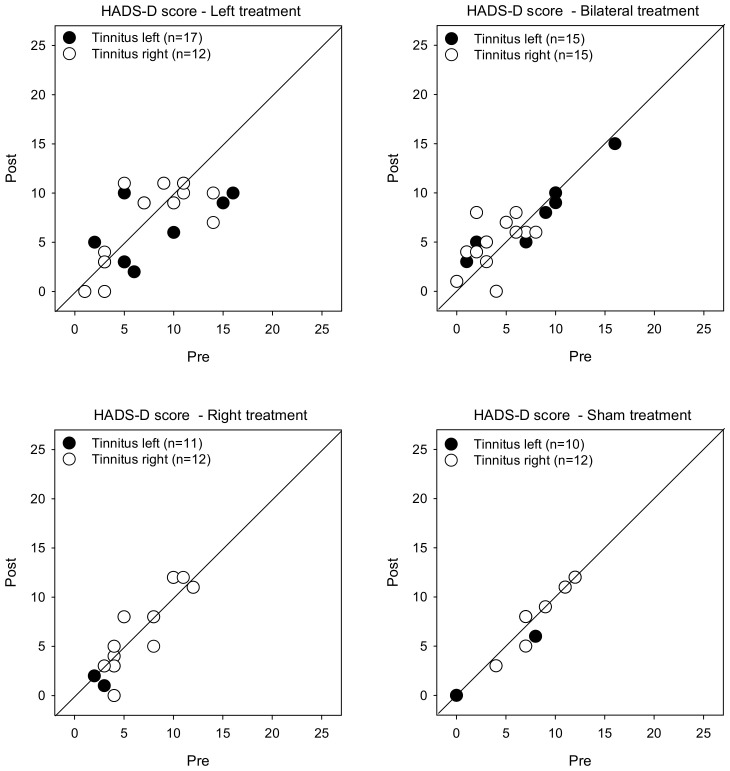
Post-treatment HADS-D scores as a function of pre-treatment scores for the Left, Right, Bilateral, and Sham treatment groups. The filled and open symbols show data for patients with left- or right-sided tinnitus, respectively. Values below the diagonal line indicate reduced depression after treatment.

**Figure 5 brainsci-12-00733-f005:**
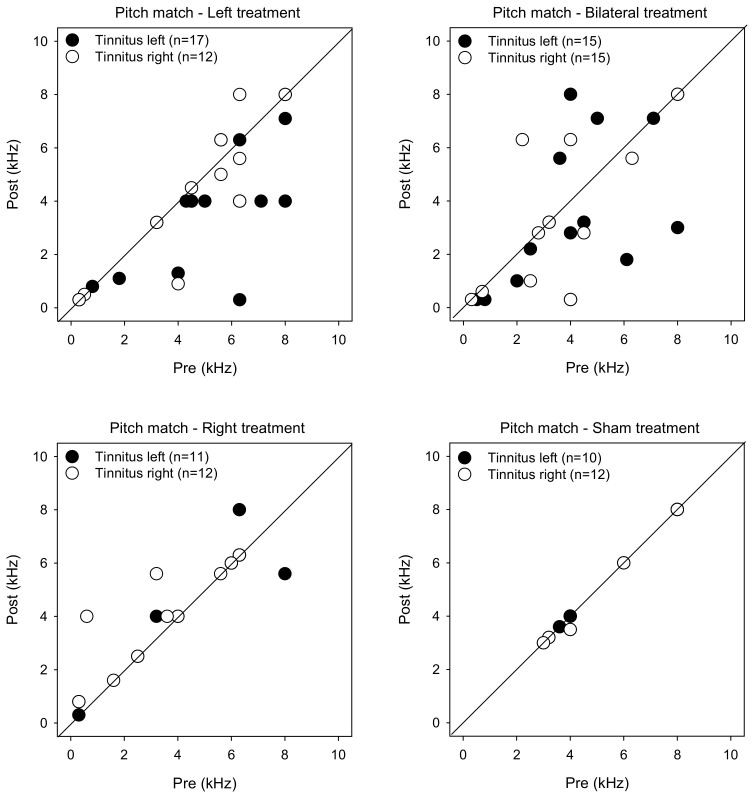
Post-treatment tinnitus pitch-matched frequency as a function of pre-treatment frequency for the Left, Right, Bilateral, and Sham treatment groups. The filled and open symbols show data for patients with left- or right-sided tinnitus, respectively. Values below the diagonal line indicate reduced tinnitus pitch frequency after treatment.

**Figure 6 brainsci-12-00733-f006:**
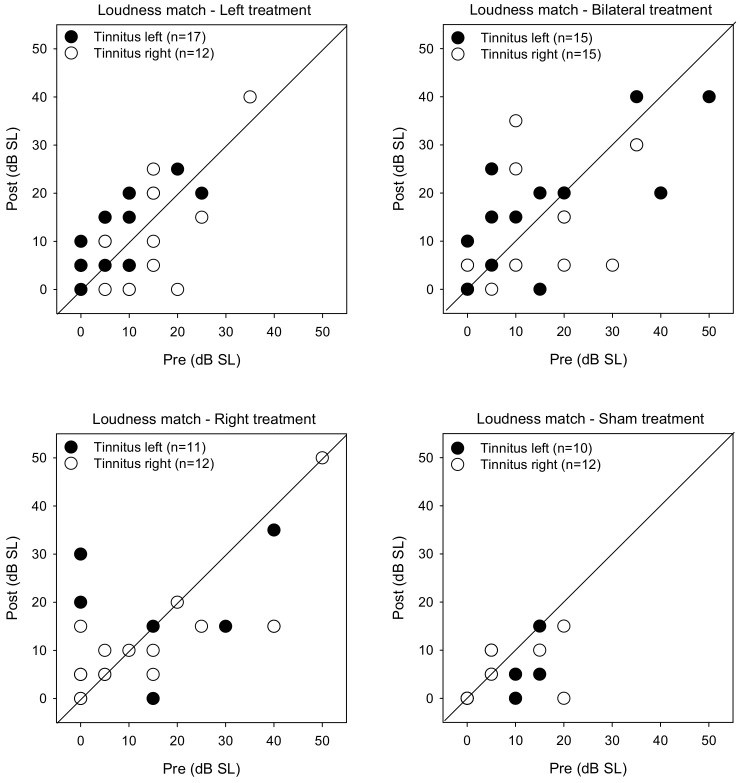
Post-treatment tinnitus loudness matching at the pitch-matched frequency as a function of pre-treatment loudness for the Left, Right, Bilateral, and Sham treatment groups. The filled and open symbols show data for patients with left- or right-sided tinnitus, respectively. Values below the diagonal line indicate reduced tinnitus loudness after treatment.

**Figure 7 brainsci-12-00733-f007:**
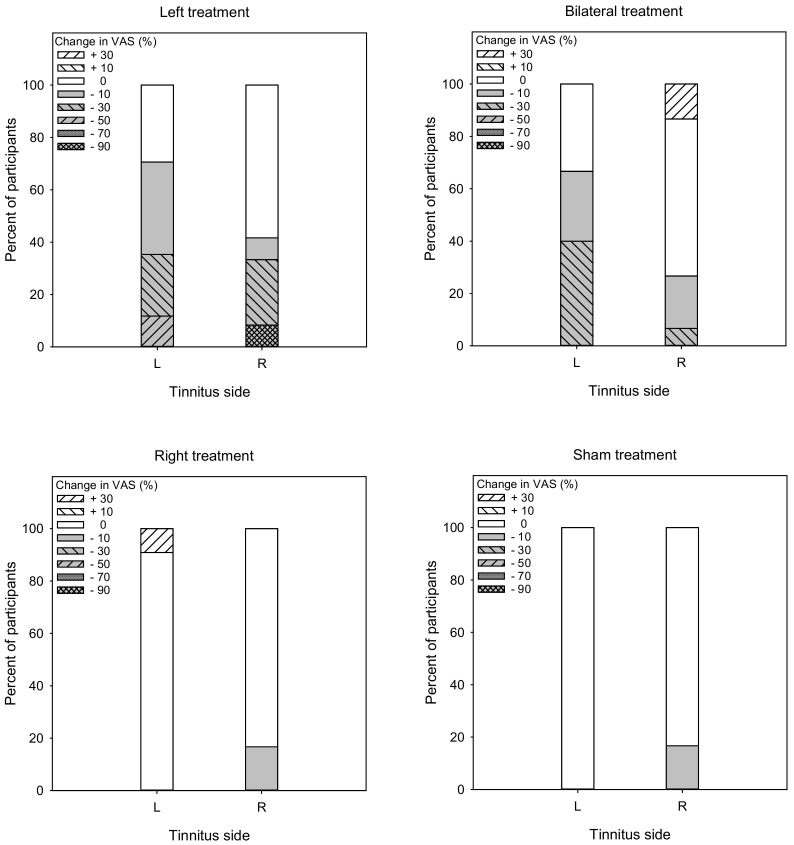
Vertical stacked bar charts showing the percentage of patients with various ranges of post-treatment changes in VAS scores the Left, Right, Bilateral, and Sham treatment groups. Reductions >10% were considered to be responders (gray bars), and reductions <10% were considered non-responders (white bars).

**Figure 8 brainsci-12-00733-f008:**
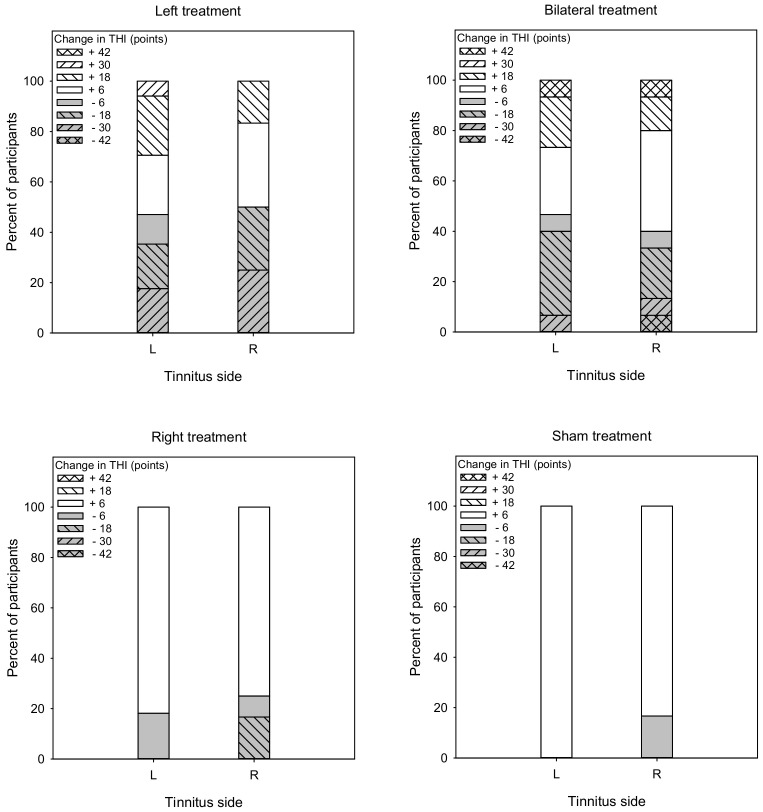
Vertical stacked bar charts showing the percentage of patients with various ranges of post-treatment changes in THI scores the Left, Right, Bilateral, and Sham treatment groups. Reductions >6 points were considered to be responders (gray bars), and reductions <6 points were considered non-responders (white bars).

**Table 1 brainsci-12-00733-t001:** Demographic information for tinnitus patients within each treatment group. Statistical analyses are shown at right. Chi-square tests were performed to compare the distributions of participant sex and tinnitus side across treatment groups. Kruskal–Wallis one-way ANOVAs were performed on ranked data to compare the distributions of age at testing, pure-tone average (PTA) thresholds in decibels hearing level (dB HL), duration of tinnitus in years, and resting motor threshold (RMT) in percent of maximum stimulator output (% MSO) across treatment groups. Significant differences according to post-hoc Dunn’s tests are shown on the far right. STD = standard deviation.

		Treatment Group						
		Left	Right	Bilateral	Sham	Statistic	*p*-Value	Post-Hoc (*p* < 0.05)
Sex (*n*)	Male	17	14	14	14	χ^2^ (3) = 1.9	0.600	
	Female	12	9	16	8			
Tinnitus side (*n*)	Left	17	11	15	10	χ^2^ (3) = 1.9	0.789	
	Right	12	12	15	12			
Age (years)	Mean	44.2	40.8	36.4	39.9	H (3) = 5.3	0.151	
	STD	13.3	10.8	11.7	10.2			
PTA (dB HL)	Mean	34.3	38.9	22.2	35.5	H (3) = 14.6	*0.002 **	Right, Sham > Bilateral
	STD	19.2	17.9	11.8	20.2			
Duration of tinnitus (years)	Mean	3.2	3.2	1.6	1.8	H (3) = 8.6	*0.034 **	Left > Bilateral
	STD	3.0	3.0	2.0	1.2			
RMT/% MSO	Mean	30.2	33.2	34.3	33.1	H (3) = 11.6	*0.009 **	Bilateral > Right
	STD	11.4	5	8.6	5.7			

Significant differences are indicated by italics and asterisks.

**Table 2 brainsci-12-00733-t002:** Mean values and standard deviation for outcome measures for each tinnitus ear (“Left ear”, “Right ear”) and across tinnitus ears (“All”) within the Left, Bilateral, and Sham treatment groups. VAS = visual analog scale; THI = Tinnitus Handicap Index; HADS-A = Hospital Anxiety and Depression Scale Anxiety subscale; HADS-D = Hospital Anxiety and Depression Scale Depression subscale; kHz = kilohertz; dB SL = decibels sensation level; STD = standard deviation.

	Tinnitus	Test		VAS	THI	HADS-	HADS-	Pitch	Loudness
	Side					A	D	(kHz)	(dB SL)
Left treatment	Left	Pre	Mean	5.8	47.4	9.4	6.9	4.9	7.9
			STD	2.0	26.8	4.4	4.4	2.4	7.3
		Post	Mean	*4.3 **	43.8	*7.9 **	6.1	*3.7 **	11.5
			STD	1.9	29.0	3.3	3.9	2.4	7.7
	Right	Pre	Mean	5.8	44.0	9.6	7.6	4.2	14.6
			STD	1.4	21.7	3.3	4.5	2.7	8.9
		Post	Mean	*4.8 **	*35.8 **	*7.4 **	7.1	3.9	11.3
			STD	2.0	24.1	2.9	4.2	2.9	12.3
	All	Pre	Mean	5.8	46.0	9.5	7.2	4.6	10.7
			STD	1.8	24.5	3.9	4.4	2.5	8.5
		Post	Mean	*4.5 **	*40.5 **	*7.7 **	6.5	*3.8 **	11.4
			STD	1.9	26.9	3.1	4	2.6	9.6
Right treatment	Left	Pre	Mean	4.9	43.2	8.7	5.4	4.5	13.2
			STD	1.8	26.6	3.4	3.2	2.6	16.3
		Post	Mean	5.0	45.9	7.5	5.2	4.5	13.2
			STD	1.8	29.7	3.0	4.0	2.5	11.7
	Right	Pre	Mean	4.7	44.4	8.1	6.4	3.8	15.4
			STD	1.9	28.0	4.0	3.2	2.2	16.2
		Post	Mean	4.5	46.2	7.3	6.2	4.4	13.3
			STD	1.9	33.3	3.4	4.0	1.9	12.9
	All	Pre	Mean	4.8	44.6	8.4	5.9	4.1	14.3
			STD	1.8	27.7	3.7	3.2	2.4	15.9
		Post	Mean	4.7	45.3	7.4	5.7	4.5	13.3
			STD	1.8	30.2	3.2	3.9	2.1	12.0
Bilateral treatment	Left	Pre	Mean	4.6	36.4	6.7	5.2	4.3	14.0
			STD	2.2	20.5	3.3	4.5	2.3	15.8
		Post	Mean	*3.7 **	35.6	5.9	6.0	3.6	14.3
			STD	2.1	24.9	2.8	3.7	2.9	13.5
	Right	Pre	Mean	4.3	38.8	7.7	5.3	3.5	17.0
			STD	2.1	23.0	3.1	4.0	2.3	12.9
		Post	Mean	4.2	36.9	*6.1 **	5.7	3.6	17.3
			STD	2.2	20.7	2.5	3.5	2.7	13.7
	All	Pre	Mean	4.5	37.6	7.2	5.2	3.9	15.5
			STD	2.1	21.5	3.2	4.2	2.3	14.3
		Post	Mean	*3.9 **	36.3	*6.0 **	5.9	3.6	15.8
			STD	2.2	22.5	2.6	3.6	2.7	13.5
Sham treatment	Left	Pre	Mean	5.0	39.6	7.4	7.6	5.5	7.5
			STD	1.3	15.2	3.6	4.5	2.2	5.9
		Post	Mean	5.0	39.6	6.6	7.4	5.5	3.5
			STD	1.3	15.7	3.3	4.5	2.2	4.7
	Right	Pre	Mean	5.7	43.0	7.7	8.3	5.4	6.3
			STD	1.9	26.8	3.3	2.8	2.2	7.7
		Post	Mean	5.5	42.7	8.2	8.0	5.3	5.0
			STD	2.1	24.8	2.0	3.3	2.3	5.6
	All	Pre	Mean	5.4	41.5	7.6	8.0	5.4	6.8
			STD	1.7	21.9	3.4	3.6	2.1	6.8
		Post	Mean	5.3	41.3	7.5	7.7	5.4	4.3
			STD	1.8	20.8	2.7	3.8	2.2	5.2

Italicized and asterisked values indicate significant differences between post- and pre-treatment measures from post-hoc Bonferroni-adjusted pairwise comparisons from LMM analyses.

**Table 3 brainsci-12-00733-t003:** Mean change and standard deviation between pre- and post-treatment outcome measures for each tinnitus ear (“Left ear”, “Right ear”) and across tinnitus ears (“All”) within the Left, Right, Bilateral, and Sham treatment groups.

			VAS	THI	HADS- A	HADS- D	Pitch (kHz)	Loudness (dB SL)
Left treatment	Left ear	Mean	−1.5	−3.6	−1.5	−0.9	−1.3	3.5
		STD	1.4	14.4	2.3	3.1	1.8	5.8
	Right ear	Mean	−0.9	−8.2	−2.2	−0.5	−0.4	−3.3
		STD	1.2	12.6	3.2	3.3	1.3	9.1
	All	Mean	−1.2	−5.5	−1.8	−0.7	−0.9	0.7
		STD	1.3	13.6	2.6	3.1	1.6	8.0
Right treatment	Left ear	Mean	0.1	1.2	1.2	−0.2	0.1	0.0
		STD	0.3	7.4	2.5	1.9	1.0	15.7
	Right ear	Mean	−0.2	0.3	−0.8	−0.3	0.6	−2.1
		STD	0.4	9.1	2.4	2.0	1.1	9.9
	All	Mean	0.0	0.7	−1.0	−0.2	0.3	−1.1
		STD	0.4	8.2	2.4	1.9	1.1	12.7
Bilateral treatment	Left ear	Mean	−0.9	−0.8	−0.8	0.8	−0.7	0.3
		STD	0.8	16.3	2.9	2.5	2.4	10.1
	Right ear	Mean	−0.1	−1.9	−1.6	0.5	0.0	0.3
		STD	0.6	18.1	2.8	2.5	1.8	13.4
	All	Mean	−0.5	−1.3	−1.2	0.6	−0.4	0.3
		STD	0.8	16.9	2.9	2.4	2.1	11.7
Sham treatment	Left ear	Mean	0.0	0.0	−0.8	−0.2	0.0	−4.0
		STD	0.0	1.3	1.7	1.0	0.0	4.4
	Right ear	Mean	−0.2	−0.3	0.5	−0.3	−0.1	−1.3
		STD	0.4	3.9	1.7	1.0	0.2	6.8
	All	Mean	−0.1	−0.2	−0.1	−0.3	0.0	−2.5
		STD	0.3	3.0	1.7	1.0	0.1	5.9

**Table 4 brainsci-12-00733-t004:** Top: Results of Pearson correlations for pre-treatment measures among age at testing, duration of tinnitus (Dur tin), VAS scores, THI scores, HADS-A scores, HADS-D scores, tinnitus pitch match, and tinnitus loudness match across all participants (*n* = 104).

	PTA		Dur tin		VAS		THI		HADS-A		HADS-D		Pitch		Loudness	
													(kHz)		(dB SL)	
	r	*p*	r	*p*	r	*p*	r	*p*	r	*p*	r	*p*	r	*p*	r	*p*
Age	0.15	0.129	0.34	*< 0.001 **	0.07	0.484	*−0.05*	0.641	−0.25	0.010	−0.09	*0.343*	0.07	0.490	−0.18	0.070
PTA			0.22	0.028	0.10	0.309	0.14	0.154	0.07	0.493	−0.05	0.583	0.17	0.080	−0.13	0.180
Dur tin					0.10	0.293	0.01	0.953	−0.03	0.769	−0.24	0.014	0.21	0.030	−0.05	0.648
VAS							0.63	*< 0.001 **	0.30	*0.002 **	0.32	*0.001 **	0.21	0.031	−0.13	0.197
THI									0.60	*< 0.001 **	0.49	*< 0.001 **	0.14	0.146	−0.01	0.950
HADS-A											0.57	*< 0.001 **	0.09	0.375	0.06	0.568
HADS-D													0.07	0.509	0.02	0.853
Pitch															−0.09	0.388

The asterisks and italics show significant relationships after Bonferroni adjustment for multiple comparisons (adjusted *p* = 0.00556).

**Table 5 brainsci-12-00733-t005:** Results of forward stepwise regression analysis to identify predictors of baseline VAS and THI scores.

	Forward Stepwise Regression (F to Enter = 4.0, *p* < 0.48)						
Dependent Variable	Variable	Coef.	Std. Coeff.	Std. Error	F-to-Remove	*p*	r^2^
VAS	Constant	2.94		0.32			
	THI	0.04	0.51	0.01	35.7	<0.001	0.26
	ANOVA: F(1, 102) = 35.7, *p* < 0.001						
THI	Constant	−12.9		4.98			
	VAS	3.8	0.30	0.89	18.0	<0.001	0.57
	HADS-A	5.1	0.60	0.60	73.0	<0.001	0.49
	ANOVA: F(1, 102) = 66.9, *p* < 0.001						

**Table 6 brainsci-12-00733-t006:** Results of forward stepwise regression analysis to identify predictors of post-treatment changes in VAS or THI scores.

	Forward Stepwise Regression (F to Enter = 4.0, *p* < 0.48)						
Dependent Variable	Variable	Coef.	Std. Coeff.	Std. Error	F-to-Remove	*p*	r^2^
VAS	Constant	0.44		0.15			
	THI	0.02	0.24	0.01	4.20	0.045	0.14
	HADS-A	0.14	0.34	0.05	8.00	0.007	0.33
	Pitch	0.28	0.47	0.07	18.40	<0.001	0.37
	ANOVA: F(3, 55) = 10.9, *p* < 0.001						
THI	Constant	0.28		2.02			
	HADS-A	2.07	0.37	0.66	9.97	0.003	0.42
	HADS-D	1.52	0.28	0.63	5.81	0.019	0.50
	ANOVA: F(1, 57) = 11.9, *p* = 0.001						

## Data Availability

The data that support the findings of this study are available on request from the corresponding author. The data are not publicly available due to privacy or ethical restrictions.

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
