# Peer review of "Effect of Ipsilateral, Contralateral or Bilateral Repetitive Transcranial Magnetic Stimulation in Patients with Lateralized Tinnitus: A Placebo-Controlled Randomized Study"

_brainsci, 2022, doi:10.3390/brainsci12060733_

Round 1

Reviewer 1 Report

Dear authors,

I have some suggestions for improvement and open questions which need to be clarified:

line 40: Why are you using the term repeated; I propose to stay with the common terminus - repetitive transcranial magnetic stimulation

line 41: where it is commonly applied is missing; I assume you mean the left temporal cortex

line 45: Which type of VAS score?

line 46: which side and protocol used is very relevant in this contex

line 57: reviews and current meta analysis could fit here or a broader picture

line 63: is there a reason why you did not include right hemispheric rTMS in your design? If I want to test the laterality of rTMS in the context of tinnitus laterality this seems to be crucial! Please state why this is missing in your design

line65-72: this information fits better in the methods part

genereal comment: Did you define any primary outcome apriori? Was the study registered in clinical trials before start? Please also improve the english a little bit before publication

line 76: Which previous treatments?

Table 1: I think that the differences between the groups should be part of your discussion; also use chi^2 for categorical variables here; which post hoc test was used?

line 97-98: I think that this is not a proper response criterion; one point on a visual analogue scale is not sufficient; please use the THI and the already published cirterions for treatment response

general comment: please also cite the instruments/ questionnaires you are using

line111: did you differentiate between noise-like and pure tone tinntus for the pitch matching? or did you only include patients with pure tone tinnitus?

line 123: left motor cortex??

line 127: Rossini method - reference is missing

line 128: what about the sham? was there another stimulator used?

line 131: how was it marked?

line 134-135: did you apply 100 repeating sequences á 10 pulses of 1hz rTMS? Why did you not use the conventional 1 hz rTMS approach? What was the intention behind this protocol with breaks?

line 142: which sham coil was used?

line 145-151: The statistical analysis secion is pretty short compared to the results you present. For the reader it is important to get the information how you analyzed your aims/ what you used for what. E.g., for what did you use the Pearson correlation. Did you check statistical assumptions?

general comment: I think that the methods as well as the results section would improve from subheadings; sections

line 159-160: what about the three treatment groups? why were they not analyzed?

Kind regards

Author Response

Dear authors, I have some suggestions for improvement and open questions which need to be clarified:

line 40: Why are you using the term repeated; I propose to stay with the common terminus - repetitive transcranial magnetic stimulation

>>Corrected

line 41: where it is commonly applied is missing; I assume you mean the left temporal cortex

>>Revised as: “Low-frequency rTMS applied to the left temporoparietal cortex…”

line 45: Which type of VAS score?

line 46: which side and protocol used is very relevant in this contex

>>In response to both comments, sentence revised as: “Marcondes et al. [12] treated 10 tinnitus patients with 1-Hz rTMS on the left temporoparietal cortex and 10 patients with sham stimulation; significant reduction in visual analog scale (VAS) scores of tinnitus severity was observed immediately after treatment only for the active rTMS group.”

line 57: reviews and current meta analysis could fit here or a broader picture

>> We have expanded this paragraph: “It is difficult to determine the benefit of rTMS beyond the placebo effect. Some studies have shown a significant benefit for real rTMS compared to sham stimulation [10,16,17], while others have not [18-20]. In a randomized controlled trial, Folmer et al. [10] observed a statistically higher response rate of tinnitus after rTMS treatment. Rossi et al. [16] com-pared active and sham rTMS in 14 patients with chronic tinnitus. After two weeks of treatment, the response rate in the active stimulation group was significantly higher than that in the sham stimulation group. In a meta review of randomized clinical trials, So-leimani et al. [17] reported a significant benefit of rTMS in terms of improved Tinnitus Handicap Inventory (THI) scores. However, other researchers have not shown a signifi-cant advantage of rTMS over sham treatment. Sahlsten et al. [20] found no significant dif-ference in the reduction of tinnitus intensity, annoyance, and stress or in THI scores be-tween active and sham stimulation, although the percent of responders was higher for the active group than for the sham group. Hoekstra et al. [18] found no significant advantage for active rTMS over sham stimulation at any time point after treatment. Collectively, these studies indicate that the benefit of rTMS for tinnitus treatment remains uncertain.”

line 63: is there a reason why you did not include right hemispheric rTMS in your design? If I want to test the laterality of rTMS in the context of tinnitus laterality this seems to be crucial! Please state why this is missing in your design

>>We apologize for the confusion. In fact, we had tested a group with Right sided treatment. We had not included this data in the original MS because there was no effect, similar to the Sham group. We have added this data back in and have updated all methods, results, figures, analyses, discussion to include these data.

line65-72: this information fits better in the methods part

>> Moved as suggested.

genereal comment: Did you define any primary outcome apriori? Was the study registered in clinical trials before start? Please also improve the english a little bit before publication

>> The study was registered with and approved by Institutional Review Board at the First Affiliated Hospital of Soochow University (#2019097). In terms of a priori outcomes, we have added to the end of the Introduction: “The primary outcome measures were tinnitus severity in terms of VAS and THI scores. Secondary outcome measures included anxiety and depression scales, pitch matches to tinnitus, and loudness matches to tinnitus.”

line 76: Which previous treatments?

>>We have modified: “…no effect of previous routine therapies (e.g., glucorticoids)…

Table 1: I think that the differences between the groups should be part of your discussion; also use chi^2 for categorical variables here; which post hoc test was used?

>>We have redone the analyses after including the Right treatment group, and now use Chi-square to analyze the participant sex and tinnitus side. Below Table 1, we have added: “While there was no significant in age at testing across groups (p = 0.151), age at testing was lower for the Bilateral (mean = 36.4 yrs; range = 18-61) than for the Left (mean = 44.2 yrs; range = 25-66), Right (mean = 40.8 yrs; range = 20-67), or Sham groups (mean = 39.9 yrs; range = 30-58). PTA thresholds were significantly lower for the Bilateral (mean = 22.2 dB HL; range = 5-50) than for the Right (mean = 38.9 dB HL; range = 10-75) or Sham groups (mean = 34.3 dB HL; range = 20-95). The duration of tinnitus was significantly longer for the Left (mean = 3.2 yrs; range = 0.08-10) than for Bilateral group (mean = 1.6 yrs; range = 0.17-7).” And later in the Discussion, we have added: “Also, the mean age at testing was lower for the Bilateral (36.4 yrs) than for the Left group (44.4 yrs), and PTA thresholds were lower for the Bilateral (22.2 dB HL) than for the Left group (34.3 dB HL), which may have contributed to the pattern of results.”

line 97-98: I think that this is not a proper response criterion; one point on a visual analogue scale is not sufficient; please use the THI and the already published cirterions for treatment response

>>We have modified the response criteria: “A response to treatment was considered when VAS scores were reduced by 10% or more [24]. And later:  “A response to treatment was considered a reduction of 6 points or more in THI scores [27,28].” We have also modified the response figures and analyses using these new criteria

general comment: please also cite the instruments/ questionnaires you are using

>>We now include references for the VAS and THI.

line111: did you differentiate between noise-like and pure tone tinntus for the pitch matching? or did you only include patients with pure tone tinnitus?

>>We have added to the Participants section: “All participants had lateralized tonal tinnitus.”

line 123: left motor cortex??

>>Revised as: “The left motor cortex M1 was stimulated using a single pulse.”

line 127: Rossini method - reference is missing

>>Reference added

line 128: what about the sham? was there another stimulator used?

>>Revised as: “For the Sham group, participants received the same treatment as the Left group, but the stimulation coil was tilted away from the skull by 45° with one wing touching the skull to induce skin sensations without inducing magnetic stimulation, as in Landgrebe et al. [19].”

line 131: how was it marked?

>>Clarified as: “…the location of the stimulation point was marked with a pen on the surface of the subject.”

line 134-135: did you apply 100 repeating sequences á 10 pulses of 1hz rTMS? Why did you not use the conventional 1 hz rTMS approach? What was the intention behind this protocol with breaks?

>>We have revised as: “For the Left group, temporo-parietal stimulation was delivered from a single coil which was positioned on the midline between T3 (left temporal midpoint) and P3 (left parietal midpoint) regions. Stimulation consisted of a train of 10 pulses (one every second), followed by two skipped pulses and then 10 more pulses, two skipped pulses, etc. The 2-second rest was implemented to reduce the possibility of epilepsy. In all, there were 100 repeating sequences totaling 1000 pulses. For the Right group, the stimulation paradigm was the same, except that the coil was positioned on the midline between T4 (right temporal midpoint) and P4 (right parietal midpoint) regions.”

line 142: which sham coil was used?

>>We have revised as: “For the Sham group, participants received the same treatment as the Left group, but the stimulation coil was tilted away from the skull by 45° with one wing touching the skull to induce skin sensations without inducing magnetic stimulation, as in Landgrebe et al. [19].

line 145-151: The statistical analysis section is pretty short compared to the results you present. For the reader it is important to get the information how you analyzed your aims/ what you used for what. E.g., for what did you use the Pearson correlation. Did you check statistical assumptions?

>>We have expanded the Data analysis section: “Data analyses were performed using IBM SPSS (version 22.0; IBM, Armonk, NY). For all analyses, significance was p < 0.05.

Demographic data were analyzed using Chi-square or one-way ANOVA, as appropriate Linear mixed model (LMM) analyses were performed to compare pre- and post-treatment outcomes across groups and across patients with left- and right-sided tinnitus. on the rTMS data. Categorical fixed effects included group (Left, Right, Bilateral, Sham), tinnitus side (left, right), and treatment (pre, post); all factors of interest were included in the analysis. Participant was a random effect (intercept) for all LMMs. A maximum likelihood model was used for the LMMs. Pairwise comparisons were performed with Bonferroni correction.

Pearson correlation analyses were used to compare pre-treatment outcome measures to one another (to observe co-linearity) and to demographic variables; Bonferroni correction was applied for multiple comparisons.

Forward stepwise regression was used to identify predictors of tinnitus severity, as well as post-treatment changes in tinnitus severity. Response rates were analyzed using Mann-Whitney analyses.”

general comment: I think that the methods as well as the results section would improve from subheadings; sections

>>We have greatly reorganized the Results and now include subheadings and sections in the Methods and Results.

line 159-160: what about the three treatment groups? why were they not analyzed?

>> We now use LMMs for the analyses. Revised as: “LMM analysis was performed on the VAS score data, with treatment group (Left, Right, Bilateral, Sham), tinnitus side (left, right), and test (pre, post) as fixed effects and patient as a random effect. A significant effect was observed for only for test [F(1, 104) = 34.0, p < 0.001], and there was a significant interaction among treatment group, tinnitus side, and test [F(3, 104) = 2.8, p < 0.045]. Post-hoc Bonferroni pairwise comparisons showed that rTMS significantly reduced VAS scores for the Left group with left- or right-sided tinnitus (p < 0.05 for both comparisons), and for the Bilateral group with left-sided tinnitus (p < 0.05). There was no significant effect of rTMS for the Right or Sham groups, and no significant effect for the Bilateral group with right-sided tinnitus.”.

And later in the analysis of the change in VAS scores, we have added: “LMM analysis was performed on the difference between pre- and post-treatment VAS scores, with treatment group and tinnitus side as fixed effects and participant as a random effect. Results showed a significant effect treatment group [F(3, 104) = 11.7, p < 0.001], but not for tinnitus side; there was a significant interaction [F(3, 104) = 2.8, p < 0.045]. Post-hoc Bonferroni pairwise comparisons showed that the reduction in VAS scores was significantly larger for the Left than for the Bilateral (p = 0.013), Right (p < 0.001), or Sham groups (p < 0.001), with no significant difference among the Bilateral, Right, and Sham groups. For the Bilateral group, the reduction in VAS scores was significantly larger for patients with left- than right-sided tinnitus (p = 0.007); there was no significant effect of tinnitus side for the Left, Right, or Sham groups. For patients with left-sided tinnitus, the reduction in VAS scores was significantly larger for the Left and Bilateral groups than for the Right and Sham groups (p < 0.05 for all comparisons), with no significant difference between the Left and Bilateral groups or between the Right and Sham groups. For patients with right-sided tinnitus, there was no significant difference in reduction of VAS scores across groups.”

Reviewer 2 Report

This study is about the comparison of treatment effect between applying site of rTMS. 

  1. Mentioned that use EEG system for anatomical references, but seems to apply same area in most patients. please explain. 
  2. Is the machine approved for tinnitus? 
  3. How do you overcome ethical issue?
  4. Please describe detail of left vs Bilateral. There is no detail for treatment time and schedule. The bilateral patients get double time compared to left treatment?
  5. Why didn't you apply right rTMS only? Bias can be made.
  6. Why the results is better in left tinnitus even in bilateral application?

Author Response

This study is about the comparison of treatment effect between applying site of rTMS. 

  1. Mentioned that use EEG system for anatomical references, but seems to apply same area in most patients. please explain. 

>>We have added to the Discussion: “Another weakness is that neither navigation nor imaging were used in this study, both of which can improve the efficacy of rTMS  [60,61]. Noh et al. [62] found that tinnitus was similarly improved by 1 Hz-rTMS delivered over the left auditory cortex when an image-guided navigation system was used or when defined as posterior to the T3-C3 line based on the 10-20 EEG System, as in Langguth et al. [30]. Sahlsten et al. [27] found no significant difference between navigated and non-navigated rTMS. While chronic tinnitus was significantly reduced in both groups, the treatment response was better in the non-navigated group in terms of reduced tinnitus intensity..”

  1. Is the machine approved for tinnitus? 

>>We have clarified: “For the Left, Right, and Bilateral treatment groups, rTMS was performed using the same CYY-I transcranial magnetic stimulator; the stimulator is approved for tinnitus treatment in China.”

  1. How do you overcome ethical issue?

>> As stated in the Participants section: “All procedures for recruitment, informed consent, and the conduct of the study adhered to the requirements of the Institutional Review Board at the First Affiliated Hospital of Soochow University (#2019097).”

  1. Please describe detail of left vs Bilateral. There is no detail for treatment time and schedule. The bilateral patients get double time compared to left treatment?

>> We have revised this section of the Methods to further clarify: “For the Left, Right, and Bilateral treatment groups, rTMS was performed using the same CYY-I transcranial magnetic stimulator; the stimulator is approved for tinnitus treatment in China. Using the International EEG System as an anatomical reference for rTMS, the location of the stimulation point was marked with a pen on the surface of the participant. The physiological condition and vital signs of the patients were closely ob-served during treatment. For the Left group, temporo-parietal stimulation was delivered from a single coil which was positioned on the midline between T3 (left temporal mid-point) and P3 (left parietal midpoint) regions. Stimulation consisted of a train of 10 bipha-sic pulses (one every second), followed by two skipped pulses and then 10 more pulses, two skipped pulses, etc. The 2-second rest was implemented to reduce the possibility of epilepsy. In all, there were 100 repeating sequences totaling 1000 pulses. For the Right group, the stimulation paradigm was the same, except that the coil was positioned on the midline between T4 (right temporal midpoint) and P4 (right parietal midpoint) regions. For the Bilateral group, temporo-parietal stimulation was simultaneously delivered from two coils that were positioned on the midline between the T3 (left temporal midpoint) and P3 (left parietal midpoint) regions and on the midline between the T4 (right temporal midpoint) and P4 (right parietal midpoint); as such there were 200 repeating sequences to-taling 2000 pulses. For the Sham group, participants received the same treatment as the Left group, but the stimulation coil was tilted away from the skull by 45° with one wing touching the skull to induce skin sensations without inducing magnetic stimulation, as in Landgrebe et al. [19]. For all groups, treatment was performed over 10 subsequent working days 5 consecutive days over a two-week period).” 

  1. Why didn't you apply right rTMS only? Bias can be made

>>As noted above, we now include data from the Right treatment group.

  1. Why the results is better in left tinnitus even in bilateral application?

>> We are unsure, but it is possible that for bilateral stimulation, stimulation on the right side inhibited the right temporoparietal cortex.

Reviewer 3 Report

The manuscript “effect of ipsilateral, contralateral or bilateral repetitive transcranial magnetic stimulation in patients with lateralized tinnitus” is a well-written manuscript that describes a placebo-controlled study of tinnitus treatment with TMS. I have only a few minor remarks on the work and recommend accepting the work after these have been addressed.

-

Page 3, Table 1: The pure-tone average hearing thresholds differ greatly between the three groups. Is there any reason for such a large difference, unlikely to have arisen by chance.

-

Page 4, line 140: Please clarify the rTMS parameters. There were pulses “for a duration of 10 s with a rest of 2 s”. Did this mean a train of 11 pulses (one every second) followed by one skipped pulse and then 11 more pulses. What was the role of the short 2 second rest between the pulses?

-

Page 4, line 143: I am not familiar with the sham coil of the used TMS device, but for other sham coils, the magnetic field is not exactly blocked from reaching the cortex. Its intensity is typically either reduced notably (but not to zero) or then its spatial shape is altered so that it stimulates other parts of the brain than the original target. See, e.g., (Smith & Peterchev, 2018). Can you clarify which type of sham coil was used here.

Smith, J.E. and Peterchev, A.V., 2018. Electric field measurement of two commercial active/sham coils for transcranial magnetic stimulation. Journal of Neural Engineering15(5), p.054001.

-

Page 5, table 2: Add units to columns “Pitch” and “Loudness”, “kHz” and I assume “dBSL”, respectively.

-

Page 10, figure 4: Also include the sham group in the figure.

-

Page 3, line 100: If the pre-treatment scores increased by treatment, should it not be counted as an adverse effect and not as a “no response”.

-

Minor notes on language

Page 3, Table 1: The units for RMT should be “RMT/% MSO”, not “RMT/mm”.

Page 3, line 122: The coil diameter was likely 92 mm and not 92 cm.

Page 3, line 126: The threshold for detecting an MEP was likely 50 µV peak-to-peak and not 50 mV in amplitude (either one or two separate mistakes, depending on if peak-to-peak or one-sided amplitude was required).

Page 4, line 170: I suggest adding “For this group, …” to sentence “there was no significant difference between pre- and post-treatment THI scores” to avoid the potential misinterpretation that there were never significant differences, and not for this group.

-

Miscellaneous minor remarks

Page 1, line 41: TMS delivers electromagnetic pulses to the brain (not just scalp).

A note on inconsistent western spelling of Chinese name compared to earlier works. Apologies for any mistakes of mine. Page 3, line 122: On published works, I found the western spelling TMS device as “Yiruide” not as “Yirider”.

Author Response

The manuscript “effect of ipsilateral, contralateral or bilateral repetitive transcranial magnetic stimulation in patients with lateralized tinnitus” is a well-written manuscript that describes a placebo-controlled study of tinnitus treatment with TMS. I have only a few minor remarks on the work and recommend accepting the work after these have been addressed.

Page 3, Table 1: The pure-tone average hearing thresholds differ greatly between the three groups. Is there any reason for such a large difference, unlikely to have arisen by chance.

>>Patients were randomly assigned to treatment groups upon enrolling in the study. Hearing status was not part of the inclusion or exclusion criteria. We have added to the Participants section: “While there was no significant in age at testing across groups (p = 0.089), age at testing was lower for the Bilateral (mean = 36.4 yrs; range = 18-61) than for the Left (mean = 44.2 yrs; range = 25-66) or Sham groups (mean = 39.9 yrs; range = 30-58). PTA thresholds were significantly lower (p = 0.002) for the Bilateral (mean = 22.2 dB HL; range = 5-50) than for the Left (mean = 34.3 dB HL; range = 10-80) or Sham groups (mean = 34.3 dB HL; range = 20-95). The duration of tinnitus was significantly longer for the Left (mean = 3.2 yrs; range = 0.08-10) than for Bilateral (mean = 1.6 yrs; range = 0.17-7) or Sham groups (mean = 1.8 yrs; range = 0.5-5).”

And later in the Discussion: “Also, the mean age at testing was lower for the Bilateral (36.4 yrs) than for the Left group (44.4 yrs), and PTA thresholds were lower for the Bilateral (22.2 dB HL) than for the Left group (34.3 dB HL), which may have contributed to the pattern of results.”

Page 4, line 140: Please clarify the rTMS parameters. There were pulses “for a duration of 10 s with a rest of 2 s”. Did this mean a train of 11 pulses (one every second) followed by one skipped pulse and then 11 more pulses. What was the role of the short 2 second rest between the pulses?

>>Revised as: “For the Left, Right, and Bilateral treatment groups, rTMS was performed using the same CYY-I transcranial magnetic stimulator; the stimulator is approved for tinnitus treatment in China. Using the International EEG System as an anatomical reference for rTMS, the location of the stimulation point was marked with a pen on the surface of the participant. The physiological condition and vital signs of the patients were closely ob-served during treatment. For the Left group, temporo-parietal stimulation was delivered from a single coil which was positioned on the midline between T3 (left temporal mid-point) and P3 (left parietal midpoint) regions. Stimulation consisted of a train of 10 bipha-sic pulses (one every second), followed by two skipped pulses and then 10 more pulses, two skipped pulses, etc. The 2-second rest was implemented to reduce the possibility of epilepsy. In all, there were 100 repeating sequences totaling 1000 pulses. For the Right group, the stimulation paradigm was the same, except that the coil was positioned on the midline between T4 (right temporal midpoint) and P4 (right parietal midpoint) regions. For the Bilateral group, temporo-parietal stimulation was simultaneously delivered from two coils that were positioned on the midline between the T3 (left temporal midpoint) and P3 (left parietal midpoint) regions and on the midline between the T4 (right temporal midpoint) and P4 (right parietal midpoint); as such there were 200 repeating sequences to-taling 2000 pulses. For the Sham group, participants received the same treatment as the Left group, but the stimulation coil was tilted away from the skull by 45° with one wing touching the skull to induce skin sensations without inducing magnetic stimulation, as in Landgrebe et al. [19]. For all groups, treatment was performed over 10 subsequent working days 5 consecutive days over a two-week period).”

Page 4, line 143: I am not familiar with the sham coil of the used TMS device, but for other sham coils, the magnetic field is not exactly blocked from reaching the cortex. Its intensity is typically either reduced notably (but not to zero) or then its spatial shape is altered so that it stimulates other parts of the brain than the original target. See, e.g., (Smith & Peterchev, 2018). Can you clarify which type of sham coil was used here. Smith, J.E. and Peterchev, A.V., 2018. Electric field measurement of two commercial active/sham coils for transcranial magnetic stimulation. Journal of Neural Engineering15(5), p.054001.

>>We have clarified: “For the Sham group, participants received the same treatment as the Left group, but the stimulation coil was tilted away from the skull by 45° with one wing touching the skull to induce skin sensations without inducing magnetic stimulation, as in Landgrebe et al. [19]. “ 

Page 5, table 2: Add units to columns “Pitch” and “Loudness”, “kHz” and I assume “dBSL”, respectively.

>>Added

Page 10, figure 4: Also include the sham group in the figure.

>>Added

Page 3, line 100: If the pre-treatment scores increased by treatment, should it not be counted as an adverse effect and not as a “no response”.

>>Revised as:  “•VAS score of tinnitus severity [23]. Patients were asked to mark their tinnitus severity on a 10-cm line, where 0 = no tinnitus and 10 = worst tinnitus imaginable. Cartoon expressions (e.g., smile, neutral, pain, extreme pain, etc.) were distributed above the line to illustrate the range of tinnitus severity. VAS scores were compared pre- and post-treatment. A response to treatment was considered when VAS scores were reduced by 10% or more [24].

• Tinnitus Handicap Inventory (THI) score [25,26]. The THI contains 25 questions with 3 response choices (Yes – 4 points; Sometimes – 2 points; No – 0 points), with a score of 100 indicating maximum tinnitus severity. A response to treatment was considered a reduction of 6 points or more in THI scores [27,28].”

Minor notes on language

Page 3, Table 1: The units for RMT should be “RMT/% MSO”, not “RMT/mm”

>>Corrected

Page 3, line 122: The coil diameter was likely 92 mm and not 92 cm.

>>Corrected

Page 3, line 126: The threshold for detecting an MEP was likely 50 µV peak-to-peak and not 50 mV in amplitude (either one or two separate mistakes, depending on if peak-to-peak or one-sided amplitude was required).

>> Revised as :“The RMT was defined as the lowest stimulator output intensity capable of inducing motor evoked potentials (MEPs) of at least 50µV (peak-to-peak amplitude) in the relaxed state for at least 5 of 10 consecutive trials [30].”

Page 4, line 170: I suggest adding “For this group, …” to sentence “there was no significant difference between pre- and post-treatment THI scores” to avoid the potential misinterpretation that there were never significant differences, and not for this group.

>>Revised as: “There was no significant effect of rTMS for the Right or Sham groups, and no significant effect for the Bilateral group with right-sided tinnitus.”

Miscellaneous minor remarks

Page 1, line 41: TMS delivers electromagnetic pulses to the brain (not just scalp)

>>Corrected as “…to the brain across the scalp…”

A note on inconsistent western spelling of Chinese name compared to earlier works. Apologies for any mistakes of mine. Page 3, line 122: On published works, I found the western spelling TMS device as “Yiruide” not as “Yirider”.

>>Corrected

Reviewer 4 Report

The study's weakness is the precision of the locality regarding the cortical area being stimulated due to the non-availability of the NBS system incorporating individual MRI of subjects. 

-Row 120-127, the coil used for stimulation was biphasic or?

- Did the authors test the handedness dominance? Row 77-86, this information is missing, it is only stated in the abstract.

-The references in the manuscript, please check if having written [8,10,16, 24,25,26,27,28,29, 30] (example row 319) it should be written [8,19,16,24-30]. Please check the whole manuscript text there are errors.

-Row 362 – “limited duration”, did the authors mean shorter duration of the tinnitus duration? Please revise the sentence to be more understandable.

-In the Discussion the author can elaborate also on the stress (which is not tested in the submitted study), not only the depression and anxiety. There are available validated scales to test also the stress related to tinnitus such as Depression Anxiety and Stress Scale 21 (DASS-21).

-Row 375, please explain to be more understandable what you mean by “improvement of mood” vs rTMS effect?

-The reference [46] Salhsten et al is not matching the reference number in the Reference list (it is [48]). Please also verify and see that this reference is [20] https://doi.org/10.1016/j.brs.2017.08.001. (2017) - Is ti reference [21] ?? Please verify entire references in the manuscript text with the reference list at the end of the manuscript. It confuses the reviewer first.

-The authors are making a comparison of their results with the Salhsten et al. but the methodology was not comparable to the present study. Salhsten et al. used the navigated rTMS, and the present study did not, so the greatest differences might be related to this fact. The authors of the submitted paper did not even comment at any spot that the greatest limitation of the present study was the fact that they could not precisely determine the location of the area that they wanted to stimulate. Salhsten et al used  navigated 1-Hz rTMS to the left superior temporal gyrus, targeted according to tonotopic representation of their individual tinnitus pitch.

Author Response

The study's weakness is the precision of the locality regarding the cortical area being stimulated due to the non-availability of the NBS system incorporating individual MRI of subjects. 

>>We have added to the last paragraph: “>>We have added to the Discussion: “Another weakness is that neither navigation nor imaging were used in this study, both of which can improve the efficacy of rTMS [60,61]. Noh et al. [62] found that tinnitus was similarly improved by 1 Hz-rTMS delivered over the left auditory cortex when an image-guided navigation system was used or when defined as posterior to the T3-C3 line based on the 10-20 EEG System, as in Langguth et al. [30]. Sahlsten et al. [27] found no significant difference between navigated and non-navigated rTMS. While chronic tinnitus was significantly reduced in both groups, the treatment response was better in the non-navigated group in terms of reduced tinnitus intensity.”

-Row 120-127, the coil used for stimulation was biphasic or?

>>We now specify biphasic pulses in the revision.

- Did the authors test the handedness dominance? Row 77-86, this information is missing, it is only stated in the abstract.

>>We did not test for handedness dominance. Note that nearly all people in China are right-handed. We have added to the end of the Participants section: “All patients were right-handed. Note that in China, the prevalence of left-handedness is less than 1% [21,22] .”

-The references in the manuscript, please check if having written [8,10,16, 24,25,26,27,28,29, 30] (example row 319) it should be written [8,19,16,24-30]. Please check the whole manuscript text there are errors.

>>We have carefully checked for errors and have conformed to the reference style.

-Row 362 – “limited duration”, did the authors mean shorter duration of the tinnitus duration? Please revise the sentence to be more understandable.

 >>Revised as “As such, the somewhat short duration of tinnitus in the present study may have increased response to rTMS.”

-In the Discussion the author can elaborate also on the stress (which is not tested in the submitted study), not only the depression and anxiety. There are available validated scales to test also the stress related to tinnitus such as Depression Anxiety and Stress Scale 21 (DASS-21).

>>We have added: “Note that stress was not tested in this study. Future studies should include instruments to measure stress in tinnitus patients such as the Depression Anxiety and Stress Scale 21 (DASS-21) [47,48].”

-Row 375, please explain to be more understandable what you mean by “improvement of mood” vs rTMS effect?

 >>Revised as: “It is possible that the therapeutic effect of rTMS observed in the present study may be relat-ed to improvement in mood (e.g., reduced HADS-A and HADS-D scores), rather than a specific rTMS effect. This is somewhat reflected by the forward linear regression analysis showing that reductions in THI scores were significantly predicted by a combination of the reduction in HADS-A and HADS-D scores.”

-The reference [46] Salhsten et al is not matching the reference number in the Reference list (it is [48]). Please also verify and see that this reference is [20] https://doi.org/10.1016/j.brs.2017.08.001. (2017) - Is ti reference [21] ?? Please verify entire references in the manuscript text with the reference list at the end of the manuscript. It confuses the reviewer first.

 >> We have carefully checked for errors and have conformed to the reference style.

-The authors are making a comparison of their results with the Salhsten et al. but the methodology was not comparable to the present study. Salhsten et al. used the navigated rTMS, and the present study did not, so the greatest differences might be related to this fact. The authors of the submitted paper did not even comment at any spot that the greatest limitation of the present study was the fact that they could not precisely determine the location of the area that they wanted to stimulate. Salhsten et al used navigated 1-Hz rTMS to the left superior temporal gyrus, targeted according to tonotopic representation of their individual tinnitus pitch.

>>We have added: “This finding is not consistent with Sahlsten et al. [49], who treated 13 tinnitus patients and found that the loudness of tinnitus decreased significantly, and the pitch of tinnitus changed in the majority of patients after rTMS treatment. Note that Sahlsten et al. [49] used navigated rTMS, while the present study did not. Navigated rTMS may have allowed for better localization of the target stimulation area. However, in a more recent study also us-ing navigated rTMS, Sahlsten et al. [27] found that the loudness of tinnitus was reduced in both the rTMS and sham treatment groups. The difference in treatment outcomes across groups was not significant, suggesting that a placebo effect may have occurred.”

And later: “Another weakness is that neither navigation nor imaging were used in this study, both of which can improve the efficacy of rTMS [60,61]. Noh et al. [62] found that tinnitus was similarly improved by 1 Hz-rTMS delivered over the left auditory cortex when an image-guided navigation system was used or when defined as posterior to the T3-C3 line based on the 10-20 EEG System, as in Langguth et al. [30]. Sahlsten et al. [27] found no significant difference between navigated and non-navigated rTMS. While chronic tinnitus was significantly reduced in both groups, the treatment response was better in the non-navigated group in terms of reduced tinnitus intensity.”

Round 2

Reviewer 4 Report

Row 501-503 in the Discussion: "While chronic tinnitus was significantly reduced in both groups, the treatment response was better in the non-navigated group in terms of reduced tinnitus intensity."

Can authors also comment on why this might happen? Probably due to stimulation intensity differences? Since if RMT is defined by navigated TMS the RMT might be defined as lower and therefore intensity used later for treatment might be lower than for the situation without using navigated TMS. Please check these data on stimulation intensity in Sahlsten et al. [27].

Author Response

Row 501-503 in the Discussion: "While chronic tinnitus was significantly reduced in both groups, the treatment response was better in the non-navigated group in terms of reduced tinnitus intensity."

Can authors also comment on why this might happen? Probably due to stimulation intensity differences? Since if RMT is defined by navigated TMS the RMT might be defined as lower and therefore intensity used later for treatment might be lower than for the situation without using navigated TMS. Please check these data on stimulation intensity in Sahlsten et al. [27].

>> In Sahlsten et al. [27], there was no significant difference (P=0.58) for resting motor thresholds between navigated group (mean=62.1%MSO) and non-navigated group (mean=59.6% MSO). The stimulation intensity was at 100% of the resting motor thresholds in both groups. The authors suggested that the target site for non-navigated rTMS may have been more optimal for tinnitus treatment. The authors referred to a study by Theodoroff et al. (2018) that showed the stimulation site according to the 10–20 EEG system was optimal target to reach the auditory cortex. The authors also suggested that the coil may have stimulated a wider brain area than the better specified target with the navigated rTMS. So we revised in text as “Sahlsten et al. [27] found no significant difference between navigated and non-navigated rTMS. While chronic tinnitus was significantly reduced in both groups, the treatment response was better in the non-navigated group in terms of reduced tinnitus intensity. The advantage with non-navigated rTMS was not due to stimulation intensity, as there was no significant difference in RMT between the navigated and non-navigated group. The authors suggested that the stimulation site according to the 10–20 EEG system may have been more optimal, stimulating a wider brain area than with the more targeted navigated rTMS.”